# Recognition and Measurement of Crypto-Assets from the Perspective of Retail Holders

**Voicu D. Dragomir *** and *Valentin Florentin Dumitru*

Faculty of Accounting and Management Information Systems, Bucharest University of Economic Studies, 6 Piața Romană, 1st District, 010374 Bucharest, Romania; valentin.dumitru@cig.ase.ro
* Correspondence: voicu.dragomir@cig.ase.ro

**Abstract:** The Markets in Crypto-Assets (MiCa) Regulation of the European Union is the first comprehensive piece of legislation that seeks to protect the interests of investors in the crypto-assets sector. Although the market value of crypto-assets is significant at world level, there is a lack of clear regulatory guidelines regarding the recognition, measurement, and presentation of crypto-assets in the financial statements of investors. Considering that not all digital assets are the same, retail holders need to take into account the characteristics, rights, and obligations associated with the crypto-assets they purchase to determine the appropriate accounting method. Therefore, the research question of the present article is: Which are the main types of crypto-assets and how should they be recognized and measured in the financial statements of investors and holders? We perform a review of the accounting policies and options, relying on relevant regulations, standards, regulatory drafts, legal and academic papers, recommendations of market regulators, crypto-asset white papers, industry opinions, and media articles. There are different accounting treatments that can be applied, depending on the legal and technological aspects of each class of crypto-assets. Based on a critical discussion of accounting policies and options, our research has implications for accounting professionals, but also for standard setters, who are urged to provide clear guidelines. Identifying the key economic characteristics of each asset and determining the most appropriate way to recognize these characteristics in the financial statements are crucial for the development of a functional and trustworthy market in crypto-assets.

**Keywords:** crypto-assets; classification; accounting; recognition; measurement; IFRS

## 1. Introduction

Crypto-assets are digital assets created on the basis of the distributed ledger technology (DLT) to validate and secure transactions [1]. They can be traded in real time while producing an immutable trace of trading activity [2]. The technology can be implemented in a centralized or decentralized crypto-exchange, depending on how crypto wallets are handled. This technology can serve a multitude of purposes, from the creation and trading of cryptocurrencies to automating contracts, issuing security-like digital tokens, guaranteeing the stable price of certain assets, or attaching a certificate of authenticity to digital art. In the legal sense, crypto-assets are "digital representations of value or rights that have potential to bring significant benefits to market participants" [3] (para. (2)). The DLT has led to the creation of a "blockchain ecosystem" [4], in which the interests of crypto issuers, investors, intermediaries, auditors, creditors, regulators, and policy-makers should converge.

Crypto-assets fulfill three distinctive economic functions: (a) serving as a means of exchange; (b) providing investment value (akin to classical art or securities); and (c) conferring economic benefits related to the participation in network arrangements and the consumption of network products and services [5]. One possible classification would be according to their liquidity. Crypto-assets can be created to be a medium of exchange

on the dedicated blockchain, with or without monetary claims against the issuer (these would be called "coins"), or to be a digital representation of rights in connection to assets, organizations, or events in the "real world" (these would be called "tokens"). Irrespective of their type, crypto-assets are powered by the DLT and rely on cryptography to verify and secure transactions on a ledger, so that their value is attributed by market participants [3] (para. 2).

Currently, there are no generally accepted standards for the recognition and measurement of crypto-assets [6]. Several countries have adopted regulations regarding cryptocurrencies, but the MiCa Regulation is the most comprehensive piece of legislation in this domain [7]. However, it does not address the recognition and measurement of crypto-assets. This is an opportunity for researchers to investigate the best solutions for increasing the usefulness of financial information pertaining to crypto-assets, especially for holders and investors (who are the vast majority of participants in the blockchain ecosystem). The IFRS Foundation, which has issued standards applied in numerous countries, has allowed companies to use professional judgment, leading to potential manipulations of earnings and financial accounts [8,9]. The strictly regulated financial reporting of crypto-assets would enhance investor confidence, reduce market risk, and promote the correct assessment of tax implications of crypto-assets.

The present article has a legalistic perspective, supported by the adoption of the Markets in Crypto-Assets (MiCa) Regulation [3]. A critical review of the literature [10] is the methodological instrument designed to summarize and assess regulatory perspectives on the accounting treatment of crypto-assets, as well as to offer recommendations to regulators and investors. Luo and Yu [11] consider that there is an urgency for researchers to offer specific guidance on the accounting treatment of established and emerging types of crypto-assets. The importance of this research is also granted by the market value of the crypto-assets. Reports show that the revenue in the cryptocurrency market is projected to reach US$37.87bn in 2023, and the yearly increase is estimated to reach 14.4% [12]. Therefore, the research question is the following: Which are the main types of crypto-assets and how should they be recognized and measured in the financial statements of holders and investors? The present article seeks to answer this question by proposing a critical review of accounting policies and options related to crypto-assets based on the International Financial Reporting Standards (IFRS).

This research is informed by the European Union (EU) regulatory framework on crypto-assets and financial accounting. The MiCa Regulation provides the main definitions and is the basis for the classification of this type of digital assets. The IFRS are mandatory for listed companies on EU stock markets and are fully compatible with the Accounting Directive 2013/34/EU [13] applicable to all companies in the EU. Moreover, the IFRS and the U.S. Generally Accepted Accounting Principles (GAAP) have a high degree of convergence [14,15]. Therefore, the present research is relevant and timely for other standard-setting efforts and markets outside the EU. From the perspective of the Organization for Economic Co-operation and Development (OECD), transparency of reporting of crypto-assets is mostly relevant for tax purposes, and its Reporting Framework [1] is compatible with the following discussion.

The article is organized as follows. The literature review methodology is presented in the next section, starting from a basic classification of crypto-assets. A presentation of the main aspects of the DLT is necessary to underpin the technological characteristics of crypto-assets. The accounting options for the recognition and measurement of crypto-assets are derived from the definitions of various types of assets based on the IFRS. Accounting policies and options are detailed for the recognition of three types of coins (cryptocurrencies, e-money tokens, and central bank digital currencies) and seven types of tokens (asset-referenced tokens, algorithmic stablecoins, security tokens, utility tokens, non-fungible tokens, hybrid tokens, and DeFi tokens). Finally, the conclusions present a summary of the proposed guidance and discuss challenges related to the recognition and measurement of crypto-assets in the financial statements of investors (specifically, companies).

## 2. Materials and Methods

The present study relies on a critical review of the literature [16] examining European legislation, accounting standards, regulatory drafts and opinions, legal and academic papers, recommendations of financial regulators, crypto-asset white papers, industry opinions, and media articles. The doctrinal aspect was underscored by using the European regulation (a primary source of law) as the benchmark for integrating academic and legal literatures. Given that the MiCa Regulation was proposed in 2020 and adopted in 2023, we rely heavily on recent sources, as recommended by van der Linden and Shirazi [17].

This analysis focuses on the IFRS and the European Union context, revealing that accounting policies and options do not arise naturally but need to be constructed as answers to changes in the economic environment [18]. As such, solutions for the recognition and measurement of crypto-assets in accounting are based on (1) the definitions of different types of assets; (2) similarities and differences between the legal aspects embedded in crypto-assets and those pertaining to typical assets; and (3) technological challenges posed by crypto-assets, such as control, commingling, the unit of account, and smart contracts. These aspects are discussed in the context of the European regulatory framework, specifically the MiCa Regulation and other relevant acts pertaining to financial instruments.

The paper proposes a conceptual framework that groups together crypto-assets based on their liquidity [19]: "coins" (or digital currencies) with the potential of being used as a means of exchange, and "tokens" as digital placeholders for rights pertaining to digital or real-world assets. As indicated above, the entire discussion is set from the perspective of investors in crypto-assets. We also provide guidance for accountants on how to recognize crypto-assets based on their technological features, encapsulated rights, and regulatory coverage. The proposed classification of crypto-assets is linked to the definitions provided in the MiCa Regulation of 2023, which is the authoritative source in the European legal environment. However, the MiCa Regulation does not provide the definition of a "token" but includes this term in the legal nomenclature.

## 3. The Distributed Ledger Technology (DLT)

"The blockchain is a technology used for structuring transactional data" [20]. It includes several nodes that store the distributed ledger. The nodes form a network where the blockchain database is located. The data stored in the database are distributed towards all the nodes of the network without needing a centralized validation [21]. The data in the ledger are replicated and synchronized in nodes in real time. The chain of blocks in the distributed ledger is maintained by the participants [22]. The database can be accessed by anyone (when it is public) or only by some specified users (when it is private). The use of the DLT in the financial industry was explored by many market participants, including crypto-assets issuers.

The blockchain technology involves the existence of three key elements [23]:

- Data: in the case of crypto-assets, the blocks that are stored on the distributed ledgers
- Network: all the nodes that work together to reach consensus
- Logic: the "smart contracts" used in transactions.

The blockchain technology has a few characteristics that solve the ownership, authenticity, and provenance issues [23]:

- It is decentralized: there is no central unit that validates the transactions. They are validated by the network nodes.
- It is immutable: in any system, the stored data are susceptible to being manipulated or modified. The node consensus makes the smart contracts immutable.
- It is secure: the data stored in the ledger are secured by cryptographically hashing each block.

Two protocols are relevant for blockchain technology: (1) the proof-of-work (PoW) and (2) the proof-of-stake (PoS) [22]. In the PoW paradigm, the digital assets are created by the mining process, which means solving a conceptual graphic algorithm called hash. After

a miner generates a new block, it will broadcast the block to the network for validation. In the PoS paradigm, network participants use smart contracts that put their digital assets at stake to operate network nodes and become validators. Just as the PoW miners receive the digital assets that they mined, the users in the PoS network receive digital assets as a reward for validating new transactions and completed blocks [22]. However, the PoS users do not always receive digital assets as a reward. They can receive, for example, lower fees for transactions. The mechanism for rewarding participants in a PoS network is dependent on the fact that validators are also stakeholders in the system [24].

To access their own digital assets and perform transactions in the blockchain network, each participant uses an alpha-numeric key, called a private key. This private key is stored in a hardware or software device called digital wallet [22]. Each wallet is secured by public and private keys and can interact with many blockchain networks, enabling its owner to send and receive crypto-assets [5]. A wallet can send and receive crypto-assets without the need for transactions to be recorded by a third party, making the transaction anonymous to anyone other than the involved parties. Each transaction is accepted by the centralized network and each new block has to comply with the cryptographic rules [25]. When a participant makes a transaction in the blockchain network, the transaction is broadcasted and the nodes that manage the network validate it [19]. The signature produced by the private key of a user represents his or her acceptance for the DLT to record the change of ownership. The miners or validators who approve this transaction receive digital assets as a fee.

Bitcoin implements the PoW protocols. The miners resolve cryptographic problems (which become more complex from one coin to the next) that are easily verifiable. Each cryptographic solution becomes a new block in the blockchain network [11]. Each new block has a reference to the previously created block, thus forming the chain, which gives the name of the technology.

The Ethereum platform (ETH) has adopted a token standard that implements an application programming interface (API) for tokens related to smart contracts. Through this API, the supply of fungible tokens can be tracked on the blockchain, and the smart contract can be executed by transferring tokens from one account to another. In contrast, the ERC-721 is a standard for non-fungible tokens [26]. In this case, smart contracts have a unique pair consisting of a contract identifier and a token identifier, similar to a file with an attached certificate of authenticity [27].

## 4. Accounting Options for the Recognition of Crypto-Assets

The definition of crypto-assets considered in the present article is based on the MiCa Regulation: "crypto-asset means a digital representation of a value or of a right that is able to be transferred and stored electronically using distributed ledger technology or similar technology" [3] (art. 3.1 (5)). This very broad definition suggests that crypto-assets can appear in different forms, including that of financial instruments [28]. However, the MiCa Regulation is not applicable to crypto-assets that may be qualified as financial instruments, deposits, funds (cash), insurance products, pension products, and social security schemes [3] (art. 2.4). Furthermore, the MiCa Regulation does not apply to non-fungible tokens of any sort [3] (art. 2.3).

EFRAG [5] enumerates the criteria for the recognition of crypto-assets in the statement of financial position. These criteria are referenced to the IFRS Conceptual Framework [29]. The following discussion is from the holder's perspective. As such, crypto-assets can be recognized as assets because they:

- Are a present economic resource controlled by the entity. Crypto-assets are a digital representation of value or contractual rights created and stored on the DLT network [5]. Cryptocurrencies and e-money tokens are the most compelling examples because they are similar to a means of exchange. Other crypto-assets correspond to the contractual right to exchange economic resources with another entity on favorable terms

(e.g., asset-referenced tokens) or rights to intellectual property (e.g., non-fungible tokens).

- Have the potential to produce economic benefits. The Conceptual framework specifies that future economic benefits do not need to be certain [29] (art. 4.14). The volatility and risks associated with crypto-assets do not affect their potential to yield economic benefits [30]. Cryptocurrencies can be sold for cash or other crypto-assets, while certain tokens can be used to receive cash or avoid cash outflows. Security tokens (which can be assimilated to ordinary stocks and bonds) can produce cash inflows through potential dividends, interest, or other capital gains.

- Are controlled by the holder entity. This is demonstrated by the holder's ability to command the use of the crypto-asset and obtain the economic benefits that may flow from it. When the crypto-asset is stored separately on a device owned by the holder (in a "cold wallet"), control is easily demonstrated [31]. However, when crypto-assets are stored in a "hot wallet" managed by a centralized exchange, the holder cannot precisely demonstrate the ability to prevent other parties from directing the use of the respective crypto-assets and obtaining economic benefits from them. For this reason, wallet providers (i.e., crypto-exchanges) may be required to ensure that user holdings of crypto-assets are kept separate from the entity's own crypto-assets [32]. In other words, the client's wallet address should be different from the custodian's wallet address, to meet the definition of control.

- Have a value that can be measured reliably. From the perspective of the holder, the purchase cost is easy to identify and can be recorded in accounting. Fair value accounting can be used for potential impairment [22].

The IFRS provides several accounting options for recognizing crypto-assets in the statement of financial position. The following enumeration is relevant for holders (not issuers, credit institutions, depositors, or intermediaries). Crypto-assets can be recognized as:

- Cash, only applicable to e-money tokens and central bank digital currencies (CBDCs). With very few exceptions, cryptocurrencies are not accepted as legal tender, and therefore they do not fall in the legal category of cash or funds. In the European Union, cash is "defined as comprising four categories: currency, bearer-negotiable instruments, commodities used as highly-liquid stores of value and certain types of prepaid cards" [33] (para. (13)). Some authors consider that cryptocurrencies should be recorded as "foreign currencies" in the financial statements [25].

- Cash equivalents, if the respective crypto-assets meet the criteria of short-term, highly liquid investments that are readily convertible to known amounts of cash, which are subject to an insignificant risk of changes in value [34]. While the most significant cryptocurrencies are highly liquid investments and readily convertible to cash, the criterion of insignificant risk of changes in value is seldom met.

- Financial instruments, if the respective crypto-assets bear ownership interest in an entity (i.e., equity) or represent contracts that impose the right to receive cash or other financial instruments from a third party [22]. Security tokens meet the definition of equity or debt instruments but are not covered by the MiCa Regulation. Additionally, tokens that encapsulate a contractual right to receive cash or a financial asset (e.g., equity) can be considered financial instruments [35] (IAS 32, art. 11).

- Inventories, when the respective crypto-assets are held for sale in the ordinary course of business [36]. IAS 2 (Inventories) was not designed to encompass crypto-assets (or non-physical assets, more generally), but the definition does not contradict the nature of crypto-assets if they are assimilated with merchandise. On the other hand, this definition excludes the investment purpose associated with some types of crypto-assets, such as security tokens.

- Prepayments, as in the case of some categories of tokens (i.e., utility, hybrid, and DeFi tokens). Prepaid expenses are usually recorded under current receivables [37].

- Intangible assets of indefinite duration, without physical substance [38]. Intangible assets are long-term assets, either to be amortized or without a limited useful life

(such as non-fungible tokens). They are usually listed in the "non-current" section of the financial statements, although the purpose of crypto-assets may contradict this classification, as they could be transformed into other assets in the short term [11].

The following sections will look in more detail at each category of crypto-assets while discussing accounting options for their recognition in financial statements. According to the principle of representing the substance of contractual rights and obligations derived from owning crypto-assets [29], the financial statements should present a true and fair view of the entity's economic resources and transactions.

## 5. Coins (Virtual Currencies)

The term "coin" refers to a crypto-asset that has the express purpose of being used as a medium of exchange [39]. The main types are listed as follows.

(a) The coins with the largest circulation are the cryptocurrencies Bitcoin and Ether. They are recorded on the DLT, not issued by any jurisdictional authority, and do not hold any claim against the issuer [5].

(b) E-money tokens are designed to maintain a stable value by referencing a single fiat currency, such as USD or EUR [17]. Issuers of e-money tokens are subject to additional constraints compared to cryptocurrency issuers, such as reserve asset custody, rules on reserve asset investment, and higher own funds requirements. E-money tokens are very similar to electronic money as defined in Directive 2009/110/EC [40]. Like prepaid bank cards, e-money tokens are electronic surrogates for coins and banknotes and are likely to be used for making payments.

(c) Central bank digital currencies (CBDCs) are cryptocurrencies designed and issued by the central monetary authority of a country, to support the cashless society and remove some of the intermediaries in the monetary system. Examples of central banks that consider issuing cryptocurrencies are the Swedish Central Bank (E-Krona), the European Central Bank (digital Euro), and the Swiss National Bank (Helvetia) [41].

For a virtual currency to function, there must be a consensus mechanism that ensures that all participants agree upon the ownership rights and transfer means [42]. The consensus mechanism at the core of the Bitcoin system is different from the stringent rules of e-money token issuers or the monetary policies of central banks that may issue CBDCs. An important aspect of this consensus mechanism is the existence of claims on the issuer. Cryptocurrencies do not hold any claims on the issuer of the coins, but many users hold these crypto-assets in virtual wallets managed by crypto-exchanges. By contrast, holders of e-money tokens should be provided with claims against the issuer—the right of redemption at par value of funds denominated in the official currency [3] (para. (19)). Moreover, the holders of CBDCs will have the strongest guarantees, considering that their coins will be legally equivalent to fiat money.

### 5.1. Cryptocurrencies

Cryptocurrencies such as Bitcoin or Ether have the following characteristics [9,18]:

(a) They are fungible crypto-assets recorded on the DLT.

(b) They do not hold an intrinsic value and are not referenced to any other asset (either cryptographic, intangible, or tangible).

(c) They are issued by private entities, not by jurisdictional authorities like central banks.

(d) They do not give the holder the right to monetary claims against the issuer or a third party.

(e) They do not give rise to a contractual right that may be settled in the holder's equity instruments.

From the perspective of financial accounting, a cryptocurrency meets the definition of intangible assets under IAS 38 because: it meets the general definition of an asset; it is identifiable and can be sold, exchanged, or transferred individually; it has no physical form; it is (generally) not legal tender [43], so it does not meet the definition of cash or

funds [39]. In this case, cryptocurrencies can be revalued at fair value through other comprehensive income [25,44,45]. However, other authors and regulatory agencies have different opinions on this matter, considering that the IFRS has not yet issued any standard or recommendation on the recognition of cryptocurrencies.

While there is a quasi-unanimous consensus that cryptocurrencies cannot be classified as cash, i.e., functional currencies [5,11,30,39], some commentators consider that cryptocurrencies can be recognized as cash equivalents under IAS 7 [45,46] because they are readily convertible and their economic substance is similar to money market instruments that are considered cash equivalents under IFRS [11,47]. Procházka [25] considers that cryptocurrencies could be presented in financial statements as cash equivalents if they are acquired in a business transaction as a medium of exchange. The IFRS Interpretations Committee [9] notes that some cryptocurrencies can be used in exchange for particular goods or services. This is a real-life option, as some companies announced in the past that some goods (e.g., Tesla cars) could be purchased with Bitcoin [48]. For cryptocurrencies recognized as cash equivalents, reporting at fair value is the adequate treatment.

Another viable option would be to recognize some cryptocurrencies as foreign currency and report the holdings at closing rate (i.e., fair value) [5]. Many authors point out that the price volatility of cryptocurrencies is the major impediment to recognizing them as cash equivalents or foreign currencies [44], but currency pairs typically exhibit volatility that may exceed 100 points per day [49]. Some cryptocurrencies are even used for hedging other commodities [50]. In either case, EFRAG [5] considers that the definition of cash and cash equivalents in IAS 7 needs to be updated to take into account the existence of different types of crypto-assets.

Cryptocurrencies can also be recognized as inventories, because IAS 2 does not require inventories to have a physical form [36]. This accounting treatment may be adequate if the entity holds cryptocurrency for sale in the ordinary course of business, i.e., in the short term [39]. This solution was also accepted by the IFRS Interpretations Committee in 2019 [9], as an alternative policy to the recognition of cryptocurrencies as intangible assets. When recognized as inventories, cryptocurrencies can only be adjusted to the minimum of their historical cost and net realizable value.

*5.2. E-Money Tokens*

Within the European Union, electronic money (e-money) is an electronic storage instrument that may be widely used to make payments to entities other than the e-money issuer [51]. From the perspective of the E-Money Directive [40] (art. 2.2), electronic money is issued on receipt of funds for the purpose of making payment transactions and is accepted by a natural or legal person other than the electronic money issuer. It is important to note that e-money represents a claim on the issuer, who shall reimburse, at any moment and at par value, without any fee, the monetary value of the electronic money held (art. 11). Similarly to electronic money, e-money tokens are "crypto-assets that aim to stabilize their value by referencing only one official currency" [3] (para. (17)). When that currency belongs to a European Union member state, the e-money tokens should be offered to the public in the Union [3] (art. 4 (5)).

E-money tokens "are electronic surrogates for coins and banknotes and are likely to be used for making payments" [3] (para. (17)). The MiCa Regulation (18) imposes two critical rules: (1) e-money tokens can only be issued by a credit institution or an electronic money institution; and (2) their holders can redeem them anytime "at par value against the currency referencing those tokens." However, e-money tokens cannot be treated as deposits [3] (para. (10)) but as cash or cash equivalents [5]. In particular, holders are not entitled to interest on holding e-money tokens [3] (para. (68)). EFRAG [5] admits that a clear definition of cash and cash-equivalents should be provided by IFRS in order to explain whether e-money can be classified as either of these [46]. This clarification will have implications for monetary policy and financial stability [5] if e-money tokens are not adequately backed by sufficient reserves in cash, in the case of financial institutions.

*5.3. Central Bank Digital Currencies (CBDCs)*

CBDCs issued using the DLT are assumed to be the "ultimate" stable asset, enabling funds to be transferred between crypto platforms [21]. Digital currencies on the blockchain would ensure a certain level of anonymity and would reduce the use of physical coins and banknotes. At the same time, retail CBDCs would be offered to the public at large, a more direct diffusion of central bank monetary policies in the economy [52]. In this respect, CBDCs would become legal tender to be used in everyday transactions. Privacy is expected to be "maximized" but not complete, so as to comply with anti-money laundering regulations [52]. Otherwise, universal access and security are expected to be the mandatory features of CBDCs to make them equivalent to legal tender [41]. CBDCs are not subject to the MiCa Regulation because they are issued by central banks and fall under other prudential rules [3] (para. (13)).

From the perspective of financial accounting, CBDCs can be defined as a new form of money, exchanged in a decentralized manner (peer-to-peer) in direct transactions between the payer and the payee [6]. Furthermore, their risk-free profile would contrast with the high volatility of cryptocurrencies. Such stable assets cannot carry a right to interest towards the holder, because their main purpose is stability, not investment. While "cash comprises cash on hand and demand deposits" [34] (art. 6), it is clear that this definition is obsolete and should be amended. On the other hand, CBDCs cannot be assimilated to cash equivalents, which are "short-term, highly liquid investments that are readily convertible to known amounts of cash" (art. 6). Considering their intended use-case as direct, anonymous, and secure money tokens with no accrued interest, CBDCs are closest to the definition of cash [6].

## 6. Tokens

The transformation of assets and rights into their digital equivalent is called "tokenization" [53]. A token can be traded whole or subdivided into fungible units, as it is a digitally transferable representation of a value or right [17,23]. Different types of contractual rights can be tokenized: rights to revenue streams, voting rights, dividend rights, ownership rights, debt-to-equity conversion rights, or rights over real-world assets [5]. Fractionalizing the ownership of real assets (such as real estate) is one benefit of tokenization. For example, the tokenization of typically illiquid assets, such as fine art, diamonds, songs, and other high-value items, provides access to a broader base of traders and increases liquidity [23]. For any type of asset, the recording of smart contracts on the DLT means that the purchase of rights cannot be erased or contested. Thus, tokenization makes markets more open and transparent. However, tokens as crypto-assets are neither issued nor guaranteed by a central bank or public authority [17].

In the present context, the term "token" refers to a crypto-asset that gives the holder additional functionality or utility [39]:

(a) Asset-referenced tokens aim to stabilize their value by referencing any combination of assets or rights, including official currencies [3] (para. (18)).

(b) Algorithmic stablecoins function on the principle of pre-programmed supply for matching asset demand, specifically for the main currencies such as USD or EUR.

(c) Security tokens are similar to equity or debt instruments, but with less intermediation and bureaucracy.

(d) Utility tokens provide access to an application or service by means of the DLT [30].

(e) Non-fungible tokens are cryptographically unique and use the blockchain to verify the validity and ownership of specific digital assets [4].

(f) Hybrid tokens are created to combine payment, utility, and investment features, with specific rights and obligations [5].

(g) DeFi tokens give access to bank-like services, such as loans, lending, and insurance, but outside the usual service channels. These tokens are exchanged on automated, decentralized platforms that operate using smart contracts.

An important criterion is fungibility. As in the case of cryptocurrencies, fungible tokens are replaceable by identical tokens issued by the same entity [5]. Most crypto-assets are fungible, which means that there is a high probability of finding an active market for the application of fair value measurement. Tokens can be virtually anything, such as reputation points, character skills in a game, lottery tickets, e-money tokens, or gold-referenced tokens [54]. Based on token fungibility and the classification of referenced assets, the following sections present each type of crypto-token that can be recognized in the financial statements of retail holders.

### 6.1. Asset-Referenced Tokens

Asset-referenced tokens [3] (art. 3.1 (6)) are also called stablecoins because they purport to maintain a stable value by referencing another value or right or a combination thereof, including one or more official currencies, a "basket of assets". Given the nature of the underlying assets, asset-referenced tokens can be on-chain collateralized stablecoins (where the underlying assets are other crypto-assets) or off-chain collateralized stablecoins (where fiat funds or commodities such as gold serve as collateral). The collateral is in the possession of the issuer or recorded on the blockchain in the wallet of the issuer [46]. Asset-referenced tokens are excluded from the definition of e-money [17].

The MiCa Regulation establishes several restrictions on the issuance and trading of asset-referenced tokens. First, holders of the asset-referenced tokens have a permanent right of redemption. This means that the issuer is required to redeem the asset-referenced tokens at any time, upon request by the holders of the respective tokens [3] (para. (57)). Redemption should always be granted in funds (other than e-money) denominated in the same currency as the purchase price of the asset, or by delivering the underlying assets (i.e., collateral). Second, asset-referenced tokens cannot carry the right to interest [55], and therefore these crypto-assets would not be used as a store of value [56]. Third, the number and value of the overall transactions with each asset-referenced token are capped at 1 million transactions or EUR 200 million in transactions per day [3] (art. 23). Thus, the MiCa Regulation imposes restrictions on stablecoins as a means of exchange to reduce the risks to financial stability [17].

Regarding off-chain collateralized stablecoins, most assets cannot be recorded or transferred without the involvement of a responsible third party. The issuer or custodian of the underlying assets will always be responsible for keeping the commodity safe in custody (outside the blockchain) and delivering the commodity when requested. Gold, silver, fiat currencies, and real estate are potential collateral for asset-referenced tokens [21]. The issuance of off-chain collateralized stablecoins is similar to the issuance of tokenized funds. The buyer/user posts eligible off-chain collateral and sends the request for new stablecoins to either the custodian or the network address specified in the smart contract. The user's funds are transformed into the eligible off-chain collateral on the market and the new stablecoins are issued by means of the smart contract. On redemption, the collateral is liquidated, and the stablecoins are bought back from the market and "burned", i.e., written-off [21].

From the perspective of financial accounting, asset-referenced tokens should be recognized as intangible assets with an indefinite useful life. As such, stablecoins are not amortized. While the impairment of asset-referenced tokens is possible under IAS 38 [38], the nature of these assets deems this procedure unnecessary. Stablecoins attempt to solve the problem of high volatility in crypto-asset markets by referencing more traditional assets [22]. In the case of on-chain collateralized stablecoins, the volatility of the underlying assets (e.g., cryptocurrencies) can undermine the very purpose of the stablecoin. In case certain stablecoins become under-collateralized, redemption is compulsory unless the buyer provides additional collateral. In accounting, providing additional collateral would result in a subsequent increase in the value of the asset, in counterparty with a decrease in the user's funds. On compulsory redemption, a penalty fee is deducted for the default

of the collateral position and the user will record a loss on the disposal of the intangible asset [21].

### 6.2. Algorithmic Stablecoins

Algorithmic stablecoins are crypto-assets that are designed to maintain price stability on the basis of a smart contract that regulates the issuance and redemption of the stablecoin to match supply and demand [19]. By design, algorithmic stablecoins are undercollateralized, meaning that there are no independent assets in reserves to back up their value [57]. There are two types of algorithms that support this type of stablecoin. Through the "rebase" algorithm, the smart contract may reduce the stablecoin supply to stabilize the price of algorithmic stablecoins when supply is in excess. In the opposite direction, the algorithm would "mint" (create) new stablecoins when demand is high, so that the price would remain constant. Tokens are minted into or burned from the holders' wallets, which contradicts the legal notion of control. An alternative model, the "seigniorage" algorithm, would facilitate a change in supply and demand between the algorithmic stablecoin and another cryptocurrency that serves as a prop [58]. In this case, the yield comes from the arbitrage trading between the algorithmic stablecoin and its cryptocurrency peg [59]. Mixed algorithmic models (called "fractional algorithmic") are also tested.

Since algorithmic stablecoins are not backed by any "real" asset (in reserves), the concept of redemption does not apply. However, this model has been criticized for its lack of credible valuation and has seen several market failures [60]. Considering that these crypto-assets are backed only by the user's expectation about the future value of their holdings, investments carry a high risk, despite being called "stable." From the perspective of financial accounting, these crypto-assets cannot be recognized as cash equivalents, because holding them carries a significant risk of devaluation. The correct accounting option would be the recognition of algorithmic stablecoins as intangible assets of indefinite duration. Furthermore, the holder should carry out impairment tests if extreme market conditions force the stablecoin to lose its peg.

### 6.3. Security Tokens

Securities are financial instruments that provide an economic stake in a legal entity. Crypto-assets have been created to mimic securities by linking the respective rights with the DLT (i.e., recording these rights on the blockchain). Security tokens can be:

(a) equity tokens—a digital representation of equity, carrying the right to vote in the general meeting of shareholders and potentially receive dividends;
(b) debt tokens—the right to principal repayment and receiving interest; or
(c) derivatives tokens—crypto-assets representing option and forward derivatives [5,46,61].

Security tokens are often sold as initial token offerings (ITO) that allow businesses to raise capital to fund a business model. The token is provided in exchange for fiat money or other crypto-assets [62]. From the perspective of the MiCa Regulation, security tokens fall within the scope of existing European Union acts on financial services [3] (para. (3)). Such tokens are fungible (just like traditional securities) and hold a claim on the issuer. For this reason, EFRAG [45] recommends treating security tokens as financial assets under IFRS 9 [63]. A token is akin to a security instrument if it represents an investment of money in an enterprise with an expectation of gains derived from the effort of the people running the enterprise [23]. For example, a crypto-asset is an equity instrument under IFRS if it embodies a right to residual interest in the net assets of a particular entity [8]. To be recognized as a financial instrument, the crypto-asset should enforce (by means of a smart contract) an agreement with clear economic consequences, such as for the holder to receive financial benefits, i.e., in the form of cash [35] (IAS 32.11 (d)), at a certain date or under certain circumstances.

Gains and losses on financial assets measured at fair value would be recognized in profit or loss, while dividends would be recognized as gains in the current period. The holder may present subsequent changes to the fair value of investments in equity through

other comprehensive income only if the respective security tokens are not held for trading purposes (in the near term). However, dividends are still recognized as gains in the current period. In any case, the MiCa Regulation [3] (art. 6.7 (c)) indicates that the prospectus of any security token should make a detailed presentation of the cash flow rights pertaining to the holder of the token, in accordance with relevant regulations on securities in public offerings [39,64].

### 6.4. Utility Tokens

Utility tokens are crypto-assets that confer any of the following rights: to access products or services on a token platform [56], purchase or sell existing or future products or services (e.g., tickets to events), engage in crypto-mining activities [17], contribute labor or resources to a system [65], program or create features of a system, decide on the functionalities of a system, get involved in a community [66], or vote on matters of governance in a token platform [5]. However, these tokens do not provide ownership or dividend rights, which would fall under the securities category [39].

From the perspective of MiCa, utility tokens are classified as "other than asset-referenced tokens and e-money tokens" [3] (para. (18)) and should not be considered as held for investment purposes [5] or as a means of payment [56]. Utility tokens are valuable if the demand for the issuer's product or service is high. Some authors consider that crypto-assets can be labeled utility tokens only if they are accepted solely by the issuer of the token [56]. However, the MiCa allows the use of utility tokens "in a limited network of merchants with contractual arrangements with the offeror" [3] (para. (26)).

From an accounting perspective, utility tokens can be classified as: prepayment assets (if they are bought to access future products or services); intangible assets (if they are held for their intrinsic value); or inventories (if they are used in the course of the business to create other products or services). Utility tokens recognized as prepayments are measured at cost and may be subject to impairment. Utility tokens recognized as intangible assets of indefinite duration can be recorded at cost and later impaired or revalued. Finally, utility tokens recognized as inventories are initially recognized at cost and subsequently measured at the lower of cost or net realizable value. Similarly, utility tokens that purport to maintain a stable value would be reclassified as asset-referenced tokens or e-money tokens, under the MiCa Regulation [56].

### 6.5. Non-Fungible Tokens (NFTs)

Non-fungible tokens (NFTs) are intrinsically linked to crypto-art [22], targeted at investors willing to invest in Ethereum (ETH) and other cryptocurrencies [67]. As their name suggests, these assets are not fungible, and thus each item is uniquely identified on the blockchain. They can be sold via centralized or decentralized crypto-exchanges. If these tokens carry other rights, such as voting or membership rights, they will be considered hybrid tokens [68]. In their simplest form, NFTs are digital artwork or other collectibles with certificates of authenticity attached to them. The certificate is a digital ID, immutably recorded using the DLT [27]. While their scarcity makes them attractive for investments [69], their innovation and disruptive impact is questionable [27]. The MiCa Regulation explicitly states that it does not apply to non-fungible tokens because the act only addresses markets in fungible crypto-assets [3] (para. (10)). From the perspective of accounting, NFTs are to be treated as intangible assets and accounted for using historical cost (indefinite useful lives) less potential impairment, but without the option of revaluation [25,44]. This solution is compliant with the prudence principle in accounting, considering that NFTs are traded on high-volatility markets [70].

### 6.6. Hybrid Tokens

Hybrid tokens can encompass various types of rights as described in the previous categories. They are created based on a tokenization process, which amalgamates crypto security features, monetary aspects, and users' incentive systems [69]. In tokenomics, each

participant is rewarded with tokens for their support of the systems (i.e., the crypto project) and continues to use tokens to redeem products or services. Hybrid tokens can have features of utility tokens or non-fungible tokens, or even resemble security tokens. The issuer can also promote such tokens as having a payment function in a limited marketplace. When the crypto-asset represents a contractual right to receive cash equivalents, it could meet the definition of a financial instrument [39]. Other important issues are how many tokens are in circulation, whether and how new tokens are added, what happens to lost tokens, how control is assessed, and what happens to expired/unsold tokens. Smart contract auditors are also an important party in this process [53].

From the perspective of financial accounting, hybrid tokens pose the difficulty of identifying the rights conveyed [22]. A token can be "a container" for a variety of rights [17]. For example, NFTs can convey the ownership right over a digital asset or physical good. If the underlying asset is physical and is used in the course of the business, then the tangible nature prevails, and the tokens could be recognized as inventories or prepayments for other assets. Thus, the rights affect the nature of the assets recorded in the accounting system. If the underlying asset has no physical form, the crypto-asset is recorded as an intangible asset with the possibility of impairment [39]. Even if the underlying asset has no physical form, the tokens can be recognized as inventories if the entity uses the crypto-assets in the ordinary course of business. It is important to understand whether the crypto-asset carries a claim on the issuer or a counterparty in addition to other rights. In this sense, the respective rights (i.e., ownership, claims, payments, equity) should be separated if they can be evaluated distinctly [5]. If the securities feature prevails, the hybrid token should comply with securities laws and be recognized as a financial instrument [19].

### 6.7. DeFi Tokens

DeFi (decentralized finance) tokens are native to a special type of digital ecosystem: a direct, peer-to-peer environment, created through decentralized applications. In this environment, users can create financial services, innovate financial instruments (which are not accessible outside the ecosystem), and tokenize financial services [71]. DeFi-focused cryptocurrencies are considered a separate type, significantly different from conventional coins [72]. They lack centralized ownership of data, control, and accountability from a single entity (e.g., a crypto-exchange or other service providers), in a logic that is not compatible with the requirements of the MiCa Regulation [41,73]. Moreover, the types of financial services offered through decentralized, trustless systems (based on smart contracts) may be contrary to the European Union regulations on financial services [59].

Overall, DeFi tokens are a disruptive innovation that has yet to be pinned down by financial regulators in the EU or elsewhere [74]. From the perspective of financial accounting, as in the case of hybrid tokens, the holder should analyze the nature of the asset and the main rights that are encapsulated in the respective tokens through one or more smart contracts [53]. The MiCa Regulation is likely to have an adverse impact on DeFi tokens because of the requirements for legal presence and accountability within the territory of the European Union [41]. Considering their characteristics, DeFi tokens can be recognized in accounting as financial instruments, prepayments, or intangible assets.

## 7. Discussion and Conclusions

The recognition of crypto-assets is partly a legal challenge, partly a technological challenge. Crypto-assets can have cash-like attributes or investment characteristics, with smart contracts embedded in the respective tokens. Their fungible or non-fungible nature can significantly influence the trading risk. The market volatility in the case of crypto-assets is significantly higher than for any other type of asset [75]. Therefore, crypto-assets are a unique type of resource whose economic and legal attributes are only approximated by the current accounting standards [5]. Table 1 summarizes the discussion presented in the current paper. The reader will notice that the same type of asset can be recognized in different ways, depending on the use case and associated rights.

**Table 1.** A summary of accounting recognition and subsequent measurement options for crypto-assets from the perspective of the holders or investors.

| Crypto-Assets | Cash | Cash Equivalents | Financial Instruments | Inventories | Prepayments | Intangible Assets |
|---|---|---|---|---|---|---|
| Cryptocurrencies * | | ☑ FV | | ☑ NR | | ☑ FV |
| E-money tokens * | ☑ IC | ☑ IC | | | | |
| CBDCs | ☑ IC | | | | | |
| Asset-referenced tokens * | | | | | | ☑ IM |
| Algorithmic stablecoins * | | | | | | ☑ IM |
| Security tokens | | | ☑ FV | | | |
| Utility tokens * | | | | ☑ NR | ☑ IP | ☑ IM |
| NFTs | | | | | | ☑ IM |
| Hybrid tokens * | | | ☑ FV | ☑ NR | ☑ IP | ☑ IM |
| DeFi tokens | | | ☑ FV | | ☑ IP | ☑ IM |

\* Subject to the MiCa Regulation. ☑ It is a recognition option under IFRS. Subsequent measurement: [IC] initial cost; [FV] fair value; [NR] minimum of the historical cost and net realizable value; [IM] initial cost (indefinite life, not amortized) less potential impairment; [IP] initial cost (short-term holding) less potential impairment.

All crypto-assets are initially recognized at cost (i.e., purchase value), but subsequent measurement may differ. For this reason, Luo and Yo [11] suggested that crypto-assets should be considered as a new asset category with a variety of characteristics and rights, technologically linked to the DLT and potentially disruptive in international markets. The discussion in our study is important for all holders of crypto-assets and users of financial statements. Thus, when crypto-assets are recognized either as cash-equivalents, inventories, or intangible assets, the impact on the asset structure of the company and the financial ratios is potentially significant.

The MiCa Regulation's goal was to impose transparency and trust requirements on crypto-assets, allowing for greater adoption of these digital products in the financial sector [17]. Therefore, crypto-assets within the scope of the MiCa Regulation need to be accompanied by a white paper detailing contractual obligations and cash flows and any potential claims on the issuer of the token. Stablecoins (i.e., asset-referenced tokens) have a claim on the issuer, which is a contractual right in the classical sense (backed by a mandatory reserve in assets). However, algorithmic stablecoins and DeFi tokens are based on smart contracts executed within a trustless system, posing a challenge to regulators. On the other hand, security tokens (with similar characteristics to equity and debt instruments) should be issued in tandem with a prospectus that specifies the actual and potential benefits, such as dividends, voting rights, or residual interest in the issuing entity.

However, the MiCa Regulation is available only in the European Union. In countries with weak money laundering regulations, the risk related to the criminal potential of cryptocurrency transactions is significant [76]. The risk is higher for cryptocurrencies with anonymous owners [75]. The cyber risk and fraud risk might appear more frequently and have a higher impact [77]. As such, in line with Ojih et al. [78] we consider that crypto-assets should be strictly regulated while not impeding technical innovation. While the MiCa Regulation aims to ensure the integrity of crypto markets, a new accounting standard on crypto-assets would protect investors and increase investor confidence [6]. Financial transparency, while against the paradigm of anonymity typical of cryptocurrencies, is convergent with the anti-money laundering efforts of the EU [79]. Financial reporting can help in that direction.

This paper contributes to accounting standard setting by providing a comprehensive classification of crypto-assets and critical insight into the recognition options for this class of assets. These items have distinctive, rapidly evolving, technologically dependent, and complex characteristics [18]. Currently, with the adoption of the MiCa Regulation, crypto-assets are not in a regulatory vacuum, but accounting standard setters still have to clarify the recognition criteria for several types of crypto-assets, as well as the recognition of revenue, cash flows, and losses pertaining to these assets [11]. The DeFi products pose even greater challenges for accounting regulators and individual holders. The composability of crypto-assets—the financial Lego [74]—is a promising feature but also a source of complexity and potential confusion for accounting professionals. The conclusion is that an update to the IFRS is long overdue [45], especially for crypto-assets, which are not classified as financial instruments.

Our paper is not free of limitations. One of them is that we have only considered the IFRS requirements when discussing the accounting treatment of crypto-assets. However, the IFRS has a worldwide applicability [80], granting relevance to our study. Further research could investigate the accounting policies and options for crypto-assets in jurisdictions (i.e., U.S.) in which other accounting standards apply (i.e., U.S. GAAP). Furthermore, the effect of the implementation of the MiCa Regulation could be studied in the future. Different perspectives on crypto-assets, such as risk management and independent auditing, need to be further investigated. We consider that different research methods (i.e., quantitative studies) could be applied to investigate the disclosure of crypto-currencies in financial statements. The accounting treatment of these assets by entities owning significant amounts is another avenue for future research. The accounting policies and options of issuers, credit institutions, depositors, and intermediaries should also be investigated.

**Author Contributions:** Conceptualization, V.D.D. and V.F.D.; methodology, V.D.D. and V.F.D.; formal analysis, V.D.D. and V.F.D.; investigation, V.D.D. and V.F.D.; writing—original draft preparation, V.D.D. and V.F.D.; writing—review and editing, V.D.D. and V.F.D. All authors have read and agreed to the published version of the manuscript.

**Funding:** This research received no external funding.

**Institutional Review Board Statement:** Not applicable.

**Informed Consent Statement:** Not applicable.

**Data Availability Statement:** Not applicable.

**Conflicts of Interest:** The authors declare no conflict of interest.

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
