# Peer review of "Recognition and Measurement of Crypto-Assets from the Perspective of Retail Holders"

_fintech, doi:10.3390/fintech2030031_

Round 1
Reviewer 1 Report
This is a topical study of the cryptocurrency markets, which assess different types of cryptocurrency assets according to how they should be included in accounting terms. I have the following comments:
- It would be interesting to include some specific examples of the types of cryptocurrency assets analysed, for instance which central banks if any use the central bank cryptocurrency?
- There needs to be some discussion on the sometimes criminal nature of cryptocurrency transactions, would this need to be considered for accounting purposes?
- In the conclusion, do the authors have any suggestions on how the cryptocurrency markets should be treated based on an international accounting standard and would this be possible?
The quality of the English language is good.
Author Response
Dear Editors and Reviewers,
We are thrilled and grateful that our paper, Classification and Recognition of Crypto-Assets from the Perspective of Retail Holders, has received so many pertinent and useful recommendations from reviewers! We did our best to respond to all recommendations and to make the paper a stronger contribution to literature! We are hopeful that the revised version will be considered closer to publication.
Below are the detailed answers to each comment.
Kind regards,
The Authors

Reviewer 2 Report
The paper contains the following sentences on page 4: 'Authors should discuss the results and how they can be interpreted from the perspective of previous studies and of the working hypotheses. The findings and their implications should be discussed in the broadest context possible. future research directions may also be highlighted'.
The presence of this sentence indicates that the authors did not proofread the article before submitting it and shows a lack of care in the preparation of the submission. I don't see why the referee should clean up the paper, as it is the authors' responsibility. Therefore, I suggest rejection.
Nothing to say.
Author Response

(The authors gave the same response as above.)

Reviewer 3 Report
Dear Authors
It was a pleasure to read your work. It is a very interesting research work that only requires minor improvements to maximize the article’s impact. Please find them in the attached file.
All the best

Author Response

(The authors gave the same response as above.)

Reviewer 4 Report
Minor commments:
Line 293 It is told that “some commentators consider that cryp-292 to currencies can be recognized as cash equivalents under IAS 7 [38,39] because they are 293 readily convertible and their economic substance is similar to money market instruments 294 that are considered cash equivalents under IFRS. Another” That should be better explained because of if the cryptocurrency bears a high risk (volatility), as it is usual, cannot be considered as cash equivalent.
Lines 554-558 Even if the underlying asset has not physical form, It can be recognized as inventories if the entity holds the crypto-asset in the ordinary course of business,
Line 514, when it says: “…but standard setter still have…” must say “…but accounting standard setter still have…”
Author Response

(The authors gave the same response as above.)

Reviewer 5 Report
This is a discussion article that analysis an emerging and fundamental issue in a digital era that is the financial reporting of crypto-assets and the need of specific regulation. For that purpose, the authors present in detail the different concepts of cryptocurrency and their financial accounting treatment as far as recognition is concerned
Table 1 gives a usefull overview of different accounting treatment for cryptocurrencies given their use and firm’s activity.
I suggest a second table that summarizes the different possibilities for measuring cryptocurrencies based on their accounting classification, namely initial measurement and subsequent measurement.
Author Response

(The authors gave the same response as above.)

Round 2
Reviewer 3 Report
In the revised manuscript, you carefully addressed the raised questions and concerns. Overall, the manuscript reads well, has clarity, and communicates your work. Congratulations on your efforts.
All the best